

# Klebsiella pneumoniae subsp. pneumoniae–bacteriophage combination from the caecal effluent of a healthy woman

Lesley Hoyles[1,2,8], James Murphy[1,8], Horst Neve[3], Knut J. Heller[3], Jane F. Turton[4], Jennifer Mahony[1], Jeremy D. Sanderson[5], Barry Hudspith[5], Glenn R. Gibson[6], Anne L. McCartney[6] and Douwe van Sinderen[1,7]

[1] School of Microbiology, University College Cork, Cork, Ireland
[2] Department of Biomedical Sciences, University of Westminster, London, United Kingdom
[3] Max Rubner-Institut (MRI), Institute of Microbiology and Biotechnology (MBT), Kiel, Germany
[4] Antimicrobial Resistance and Healthcare Associated Infections Reference Unit, Public Health England–Colindale, London, United Kingdom
[5] Department of Gastroenterology, Guy's and St Thomas' NHS Foundation Trust and King's College London, London, United Kingdom
[6] Food Microbial Sciences Unit, Department of Food and Nutritional Sciences, University of Reading, Reading, Berkshire, United Kingdom
[7] Alimentary Pharmabiotic Centre, University College Cork, Cork, Ireland
[8] These authors contributed equally to this work.

Corresponding authors
Lesley Hoyles,
l.hoyles@westminster.ac.uk
Douwe van Sinderen,
d.vansinderen@ucc.ie

## ABSTRACT

A sample of caecal effluent was obtained from a female patient who had undergone a routine colonoscopic examination. Bacteria were isolated anaerobically from the sample, and screened against the remaining filtered caecal effluent in an attempt to isolate bacteriophages (phages). A lytic phage, named KLPN1, was isolated on a strain identified as *Klebsiella pneumoniae* subsp. *pneumoniae* (capsular type K2, *rmpA*+). This *Siphoviridae* phage presents a rosette-like tail tip and exhibits depolymerase activity, as demonstrated by the formation of plaque-surrounding haloes that increased in size over the course of incubation. When screened against a panel of clinical isolates of *K. pneumoniae* subsp. *pneumoniae*, phage KLPN1 was shown to infect and lyse capsular type K2 strains, though it did not exhibit depolymerase activity on such hosts. The genome of KLPN1 was determined to be 49,037 bp (50.53 %GC) in length, encompassing 73 predicted ORFs, of which 23 represented genes associated with structure, host recognition, packaging, DNA replication and cell lysis. On the basis of sequence analyses, phages KLPN1 (GenBank: KR262148) and 1513 (a member of the family *Siphoviridae*, GenBank: KP658157) were found to be two new members of the genus "Kp36likevirus."

## INTRODUCTION

Metagenomic studies on samples from a range of different environments and the potential of bacteriophage (phage) therapy to treat antibiotic-resistant, clinically relevant bacteria have renewed interest in virus-like particles (VLPs). Metagenomic (virome) studies, in particular, have demonstrated that VLPs are the most genetically diverse entities in the biosphere (*Reyes et al., 2012*). Given their staggering abundance and diversity, coupled to their perceived crucial role in the functioning of ecosystems, it is surprising that VLPs (and by extension, phages) remain the most poorly characterized biological entities (*Reyes et al., 2012*).

Virome and classical studies examining VLPs in the faeces of adults and infants have demonstrated that there is a vast diversity and abundance of phages associated with the human gut microbiota (*Breitbart et al., 2003*; *Breitbart et al., 2008*; *Lepage et al., 2008*; *Reyes et al., 2010*; *Kim et al., 2011*; *Minot et al., 2011*; *Wagner et al., 2013*; *Hoyles et al., 2014*). Similar to the prokaryotic make-up of the human gut microbiota, each individual harbours a unique virome (dsDNA, ssDNA, RNA) that is temporally stable, and whose composition appears to be influenced by diet (*Reyes et al., 2010*; *Minot et al., 2011*; *Minot et al., 2013*). *Lepage et al. (2008)* reported that mucosal biopsies taken from healthy individuals and Crohn's patients contained between $4.4 \times 10^7$ and $1.7 \times 10^{10}$ (mean $1.2 \times 10^9$) VLPs/biopsy, with Crohn's patients harbouring significantly more VLPs in their biopsies than healthy individuals. A recent study in which phage populations in human faecal and caecal-effluent samples were estimated using epifluorescence microscopy (EFM) and transmission electron microscopy (TEM) reported the presence of up to $10^{12}$ VLPs/g faeces (EFM) and at least $1 \times 10^5$ VLPs/ml caecal effluent (TEM) (*Hoyles et al., 2014*). However, little is known about the host ranges of these VLPs, and their abundance within other regions of the gastrointestinal tract.

Generation of comprehensive gut virome data has been hampered by the lack of available methods for concentrating VLPs within faecal samples and extracting sufficient amounts of DNA from them so sequencing can be performed without the need to amplify the template DNA (*Reyes et al., 2012*). However, the recent publication of a PEGylation method applied to human faecal samples and demonstration that microgram quantities of DNA can be isolated from faecal VLPs has gone some way to overcome this substantial technical hurdle (*Hoyles et al., 2014*). When it comes to annotation of virome sequence data, a major stumbling block is the lack of available phage genome sequences against which contigs can be compared, plus the large number of viral genes that are currently not represented in sequence databases. For example, studies of the human faecal virome have reported that between 66% and 98% of the generated sequences have no significant hits with GenBank sequences (*Breitbart et al., 2008*; *Reyes et al., 2010*; *Minot et al., 2011*). There are few, if any, available sequences for human-gut-associated lytic phages so there is a need to isolate and genomically characterize phages from gastrointestinal sources, rather than relying on the assumption that phages found in sewage are primarily of human gut origin. However, sequences of (predicted) prophages from genome sequences of bifidobacteria, lactobacilli, *Helicobacter pylori* and *Escherichia coli* (strain Nissle 1917)

of gastrointestinal origin are available (*Ventura et al., 2005*; *Villion & Moineau, 2009*; *Vejborg et al., 2010*; *Luo et al., 2012*).

During a study of the microbiota associated with the caecum of patients with Irritable Bowel Syndrome and healthy controls, attempts were made to isolate bacterium–phage combinations from samples of caecal effluent. Herein, we report the isolation and characterization of a *Klebsiella pneumoniae* subsp. *pneumoniae*–phage combination from the caecal effluent of a healthy woman, and the implications of our findings in relation to gastrointestinal microbial ecology and the potential of the human gut as a source of phages with therapeutic uses. Previous studies describing the isolation of lytic phages against *K. pneumoniae* have reported recovery of these entities from wastewater, seawater, sewage and sewage-contaminated water samples, but never directly from gastrointestinal contents (*Souza, Ginoza & Haight, 1972*; *Kumari, Harjai & Chhibber, 2009*; *Kumari, Harjai & Chhibber, 2010a*; *Drulis-Kawa et al., 2011*; *Hung et al., 2011*; *Cui et al., 2012*; *Hsu et al., 2013*; *Karumidze et al., 2013*; *Jamal et al., 2015*).

## MATERIALS AND METHODS

### Ethical approval and patient information

Ethical approval to collect caecal effluent from patients was obtained from St Thomas' Hospital Research Ethics Committee (06/Q0702/74) covering Guy's and St Thomas' Hospitals, and transferred by agreement to London Bridge Hospital. Patients provided written consent to provide samples. The caecal effluent was collected as described by *Hoyles et al. (2014)* from a 31-year-old female who showed no evidence of colonic abnormalities or disease as based on a routine colonoscopic examination.

### Sample processing, and isolation of bacteria and phages from caecal effluent

The sample was transported to the University of Reading under anaerobic conditions (in a gas jar with an anaerobic gas-generating pack; Oxoid Ltd, Hampshire, UK) and on ice, where it was processed within 3 h of collection. The sample was transferred to an anaerobic cabinet (Whitley MG1000 anaerobic workstation; DW Scientific, West Yorkshire, UK; gas composition 80% $N_2$, 10% $H_2$, 10% $CO_2$) and mixed well by shaking. An aliquot (1 ml) of the sample was diluted with 9 ml sterile, anaerobic half-strength peptone water (Oxoid) in a sterile universal bottle containing 2-mm glass beads. A dilution series ($10^{-1}$ to $10^{-6}$) was prepared from the homogenate in sterile, anaerobic half-strength peptone water (Oxoid). Aliquots (20 μl) were plated in triplicate onto fastidious anaerobic agar (BIOTEC laboratories, Ipswich, UK) containing 5% laked horse blood. Bacteria were incubated anaerobically for 5 days at 37 °C, and then enumerated. Ten colonies were selected randomly, streaked to purity, and stored on Microbank cryogenic beads (Prolab Diagnostics, Merseyside, UK) at −70 °C.

The remaining neat caecal effluent (∼25 ml) was processed as described by *Hoyles et al. (2014)*. Briefly, the sample was diluted 1:4 (v/v) with sterile TBT buffer (100 mM Tris/HCl, pH 8.0; 100 mM NaCl; 10 mM $MgCl_2 \cdot 6H_2O$ filtered through a 0.1 mm pore size filter prior to autoclaving). The sample was homogenized in a stomacher (Stomacher

400 Lab System; Seward, West Sussex, UK) for 2 min at 'high' speed, and then placed on ice for 2 h to allow detachment of phages from and settlement of particulate material. The homogenate was centrifuged at 11,180 **g** for 30 min at 4 °C, and the supernatant passed through a sterile 0.45 μm cellulose acetate filter (Millipore, Billerica, Massachusetts, USA). The filtrate was stored at 4 °C until used in spot assays.

## Identification of isolated bacteria

DNA was isolated from bacteria using InstaGene Matrix (Bio-Rad, Deeside, UK) according to the manufacturer's instructions. Partial (∼600 nt) 16S rRNA gene sequences were obtained for the isolates via the University of Reading's Biocentre. Nearest relatives of isolates were determined using EzTaxon (*Chun et al., 2007*). The identity of strain L4-FAA5, tentatively identified as *Klebsiella pneumoniae* on the basis of its 16S rRNA gene sequence, was confirmed using a species-specific PCR that detects the 16S–23S internal transcribed spacer unit of *K. pneumoniae* as well as capsular-type-specific, and virulence gene targets (*Turton et al., 2010*). Malonate and Voges–Proskauer reactions were positive indicating that the isolate was subsp. *pneumoniae* (rather than subsp. *ozaenae*) (*Turton et al., 2010*). Typing was carried out by VNTR analysis at loci A, E, H, J, K and D (*Turton et al., 2010*) and an additional three loci (N1, N2 and N4). Primers, repeat sizes and flanking sequence sizes for the additional loci were: N1F 5'-CATCAGGTGCAAGATTCCA-3' and N1R 5'- TGAGCGATTGCTGGCCTA-3', 116 bp repeat with a 107 bp flanking sequence; N2F 5'-GATGCGGCAAGCACCAC-3' and N2R 5'-ACGCCCTGACCATTATGC-3', 57 bp repeat with a 109 bp flanking sequence; and N4F 5'-GTGCGGTGATTGTGATGG-3' and N4R 5'-CTGACAACGTCGATGTGG-3', 67 bp repeat with a 119 bp flanking sequence.

## Screening of bacteria against filtered caecal effluent

*K. pneumoniae* strains isolated from the caecum were grown to $OD_{660} \sim 0.4$ in tryptone soya broth (Oxoid Ltd), and used in spot assays as follows. An aliquot of culture (200 μl) was inoculated into 3 ml tryptone soya broth containing 0.3% (w/v) agarose (SeaKem LE agarose; Lonza Rockland) that had been heat-treated by microwaving and dispensed aseptically from a larger volume before cooling to 48 °C. The overlays containing bacteria were poured over 20 ml solid agar plates of autoclaved tryptone soya agar (Oxoid Ltd). Once the agar had solidified, a 10 μl spot of filtered caecal effluent was applied to the overlays, and the plates were incubated overnight at 37 °C in the anaerobic cabinet. Identical plaques were observed on all *K. pneumoniae* isolates; strain L4-FAA5 was used to propagate and purify an isolated phage in tryptone soya broth or reinforced clostridial and nutrient media (Oxoid Ltd). Neither calcium nor magnesium was added to media during propagations.

The ability of the purified phage to infect a panel of *K. pneumoniae* subsp. *pneumoniae* clinical isolates (Table 1) was determined using the spot assay as described above, except that nutrient agar plates were used for the base. Strain L4-FAA5 was used as a positive control.

**Table 1 Description of *K. pneumoniae* subsp. *pneumoniae* isolates against which phage KLPN1 was screened.**

| Strain[a] | Capsular type (K PCR result)[b] | *rmpA* | *wcaG* | Source | Infected by phage KLPN1 |
|---|---|---|---|---|---|
| L4-FAA5 | K2 | + | − | Human caecal effluent | Yes |
| K/5216 | K1 (K1 cluster of CC23) | + | + | Liver abscess (Taiwan) | No |
| NCTC 5055 | K2 (reference strain) | + | − | Human | Yes[d] |
| NCTC 9660 | K5 (reference strain) | − | − | Cloacae of horse | No |
| PHE1 | − | − | − | Rectal swab | No |
| PHE2 | − | − | − | Human clinical | No |
| PHE3 | − | − | − | Sputum, transplant patient | No |
| PHE4 | − | − | − | Urine, spinal injury patient | No |
| PHE5 | − | − | − | Human blood | No |
| PHE6 | − | − | + | Urine, incontinent patient | No |
| PHE7 | − | − | − | Human clinical | No |
| PHE8 | − | − | − | Human blood | No |
| PHE9 | − | − | − | Human clinical | No |
| PHE10 | − | − | − | Human blood | No |
| PHE11[c] | K2 | − | − | Blood, patient with urinary tract infection | Yes[d] |
| PHE12[c] | K2 | − | − | Urine | Yes[d] |
| PHE13 | K2 | − | − | Blood and sputum, patient with bacteraemia and pneumonia | Yes[d] |
| PHE14 | K2 | + | − | Sputum, patient with bacteraemia | Yes[d] |
| PHE15 | K2 | − | − | Urine, cardiac patient | Yes[d] |
| PHE16 | K20 | + | − | Sputum, transplant patient | No |
| PHE17 | K54 | − | + | Intensive care unit | No |
| PHE18 | K57 | − | − | Sputum, transplant patient | No |

**Notes.**

nd, No data.

[a] Strains with the prefix PHE were submitted for typing by healthcare providers to Public Health England–Colindale. Each isolate represented a distinct strain, with the exception of isolates PHE11 and PHE12.

[b] K PCR can detect K1, K2, K5, K20, K54 and K57 capsular types.

[c] Corresponds to multi-locus sequence type ST14, often seen among multi-drug-resistant isolates producing carbapenemases.

[d] Phage KLPN1 did not exhibit depolymerase activity on these K2 isolates, but it did on L4-FAA5.

## Purification of phage particles

A 100-ml culture of strain L4-FAA5 was grown to mid-exponential phase, inoculated with 100 µl of phage stock ($\sim$10$^{10}$ pfu/ml) and incubated until the medium became clear ($\sim$2 h after infection). The lysate was centrifuged at 4,500 $g$ for 10 min, then passed through a 0.45 µm cellulose acetate filter. PEG 8000 (10%, w/v) and NaCl (6%, w/v) were added to the filtrate, and mixed until all particulates had dissolved. The sample was left at 4 °C for 16 h, then centrifuged at 4,500 $g$ for 30 min. The supernatant was removed and the pellet resuspended in 4 ml TBT buffer. A CsCl block gradient was formed with 5 M and 3 M CsCl, both prepared in TBT buffer and samples were placed on top of this gradient and subjected to centrifugation at 100,000 $g$ for 2 h at 4 °C. The band containing $\sim$2 ml of TBT buffer and the phages was drawn out of the tube and the purified sample was transferred to dialysis tubing (4,000–6,000 Da cut-off), and dialysed against TBT buffer overnight. The

TBT buffer was replaced with fresh TBT buffer, and the sample dialysed for a further 4 h. The purified phage particles were removed from the dialysis tubing and stored at 4 °C.

## Transmission electron microscopy

This was done as described by *Hoyles et al. (2014)*.

## Extraction of DNA from phage KLPN1 and restriction enzyme profiles

An aliquot (250 μl) of the purified phage particle stock was treated with DNAse (2 U) for 30 min at 37 °C, then heated for 10 min at 80 °C. Two phenol/chloroform extractions were performed, before the DNA was precipitated using 1/10 volume of 3 M sodium acetate (pH 4.8) and 2 volumes of ice-cold ethanol. After air-drying, DNA was resuspended in 100 μl TE buffer. Restriction profiles were obtained for phage KLPN1 using the enzymes EcoRV and HaeIII (SuRE/Cut; Roche Applied Science, Basel, Switzerland): restriction digests contained 3 μl $H_2O$, 1 μl enzyme, 1 μl enzyme buffer and 5 μl DNA, and were incubated for 3 h at 37 °C before being run on a 0.8% (w/v) agarose gel (1 h, 75 V) and stained with ethidium bromide.

## Whole-genome sequencing and annotation of the phage genome

DNA (5 μg) was extracted and concentration verified by Nanodrop quantification. Confirmatory molecular identification tests were also conducted on the DNA extract prior to shipment to the contract sequencing facility (Macrogen Inc., Seoul, Korea). Eighty-fold sequencing coverage was obtained using pyrosequencing technology on a 454 FLX instrument. The individual sequence files generated by the 454 FLX instrument were assembled with GSassembler (454 Lifesciences, Branford, Connecticut, USA) to generate a consensus sequence. To ensure correct assembly and resolve any remaining base-conflicts, short segments of the genome were amplified by PCR and the generated amplicons were then subjected to Sanger sequencing (MWG, Ebersberg, Germany). Open reading frames (ORFs) were automatically predicted using Genemark (*Besemer & Borodovsky, 1999*). General feature format (gff) files were generated for the predicted phage proteome (retaining proteins with a minimum size of 30 amino acids) and visualized using the annotation software Artemis v10.0 (*Rutherford et al., 2000*). ORF boundaries were verified and, where required, adjusted by manual inspection of Shine–Delgarno sequences. BLASTP (*Altschul et al., 1990*) was used to provide preliminary functional annotation data, and to carry out a comparative analysis of KLPN1 with previously sequenced *Klebsiella* phages at the protein level (*Altschul et al., 1990*). To improve genome annotations/predictions, all proteins encoded by ORFs in the genome sequence of KLPN1 were searched against InterProScan (http://www.ebi.ac.uk/interpro/). HHpred (http://toolkit.tuebingen.mpg.de/hhpred) searches were done to identify domains associated with depolymerase activity.

## Screening virome datasets for phage sequences related to KLPN1

Fasta files associated with public virome datasets were downloaded from the METAVIR web server (http://metavir-meb.univ-bpclermont.fr; *Roux et al., 2011*) on 31 March 2015

(Table S1). Each fasta file was used to create a BLAST database, against which the genome sequence of phage KLPN1 was searched using BLASTN.

## RESULTS

### Isolation and characterization of bacteria isolated from caecal effluent

On fastidious anaerobe agar, $2.22 \times 10^8 \pm 5.30 \times 10^7$ ($n = 3$ replica plating) bacteria were isolated per millilitre of caecal effluent sample ($\log_{10} 8.35 \pm 7.72$ bacteria/ml caecal effluent). Ten colonies were randomly selected from one of the triplicate plates used to enumerate the bacteria, and streaked to purity. On the basis of 16S rRNA gene sequence analysis (EzTaxon), isolates represented *Klebsiella pneumoniae* ($n = 5$), *Bacteroides vulgatus* ($n = 2$), *Bacteroides massiliensis* ($n = 1$), a novel member of the order *Erysipelotrichales* distantly related to [*Clostridium*] *innocuum* ($n = 1$), and *Haemophilus parainfluenzae* ($n = 1$).

It is widely accepted that identification of members of the family *Enterobacteriaceae* is difficult on the basis of 16S rRNA gene sequence data alone; therefore, one of the strains identified as *K. pneumoniae*, L4-FAA5, was characterized using the PCR-based methods of *Turton et al. (2010)*. The isolate was found to be a strain of *K. pneumoniae* subsp. *pneumoniae*, capsular type K2, $rmpA^+$. Its VNTR profile was unique among the clinical isolates tested and on the wider database of clinical isolates held by Public Health England–Colindale.

### Screening of filter-sterilized caecal effluent against bacteria to isolate phages

The *K. pneumoniae* isolates were used in spot assays with the caecum filtrate, which was free of bacteria (as verified by Gram-stained smear and plating on nutrient agar; not shown). Phages infecting these isolates were initially identified in the caecal effluent by spotting 10 µl aliquots of filtered caecal effluent onto TSA overlays containing 200 µl of a given culture in the exponential phase of growth. Plaques identical to those shown in Fig. 1A were observed with all *K. pneumoniae* isolates in the spot assays, and phages were present at $2 \times 10^5 \pm 2.65 \times 10^3$ ($n = 3$) pfu/ml caecal effluent. Clear plaques of 2 mm diameter were visible within 3 h of spotting onto an agar overlay. After prolonged incubation, the area around the plaques developed opaque haloes presumably caused by depolymerase activity, which increased in size over the course of 4 days, although the central clear area of the plaques remained 2 mm in diameter (Fig. 1B). This phenomenon has been observed previously for lytic phages of the families *Siphoviridae*, *Podoviridae* and *Myoviridae* against *K. pneumoniae*, and is associated with degradation of capsular polysaccharides (*Geyer et al., 1983*; *Verma, Harjai & Chhibber, 2009*; *Hsu et al., 2013*; *Lin et al., 2014a*; *Shang et al., 2015*). The non-*Klebsiella* isolates were not screened against the caecum filtrate for phages.

Strain L4-FAA5 was used as the host bacterium on which to isolate and propagate one phage (which we named KLPN1) to purity. KLPN1 infected all *K. pneumoniae* isolates recovered from the filter-sterilized caecal effluent.

The ability of phage KLPN1 to infect a panel of *K. pneumoniae* subsp. *pneumoniae* clinical isolates was determined (Table 1), which revealed that all K2 strains tested

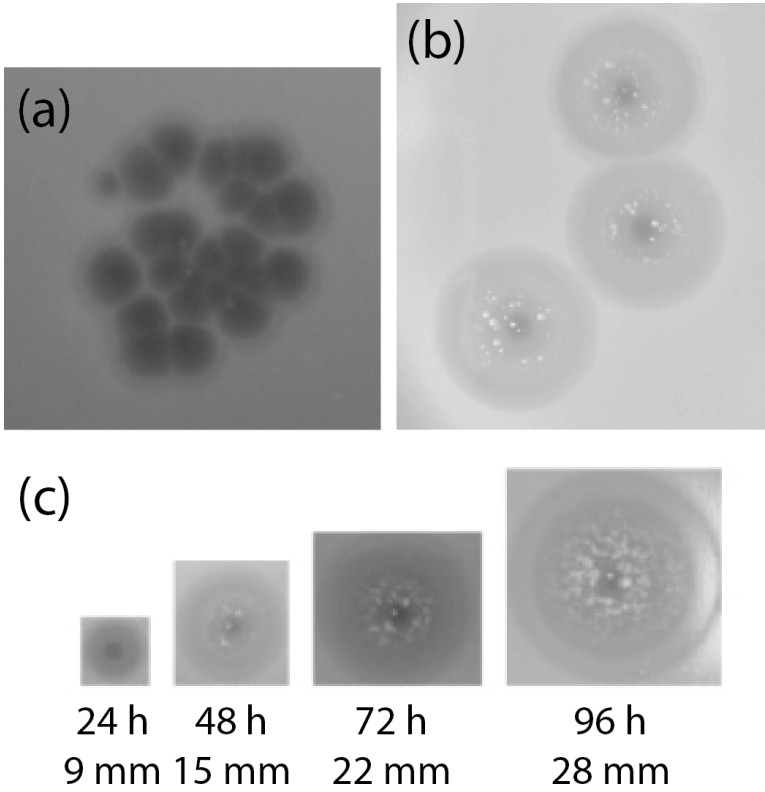

**Figure 1 Appearance of plaques formed on *K. pneumoniae* subsp. *pneumoniae* L4-FAA5 by phage KLPN1.** (A) Initial isolation of phages from filtered caecal effluent on TSA. (B) Appearance of plaques of pure phage stock after 24 h. (C) Growth of haloes surrounding plaques over the course of the course of 96 h. After 48 h, presumed phage-insensitive mutants can be seen growing in the haloes surrounding the plaques. Images are shown to scale.

(including the type strain) were sensitive to this phage, though KLPN1 did not infect any of the other tested strains belonging to other capsular types. VNTR analysis showed that the six K2 isolates in the panel represented five, distinct strains, suggesting a wide susceptibility to the phage among isolates of this capsular type. In addition, it did not exhibit depolymerase activity with any of the clinical K2 isolates.

## Characterization of phage KLPN1

Phage KLPN1 is chloroform-resistant. In addition, it displayed stability to prolonged storage in TSB at 4 °C: after 6 and 18 months' storage, titres for the phage were still $10^{10}$ pfu/ml, comparable with the original stock. Transmission electron microscopy revealed the isometric-headed phage to possess a capsid of ∼62.7 ± 2.3 nm ($n = 30$) in diameter with a long non-contractile tail of ∼164.4 ± 3.0 nm ($n = 29$), thus indicating that this phage is a member of the family *Siphoviridae* (Fig. 2). Notably, the base-plate structure was unusual with a distinct central tail fibre [length 33.4 ± 1.7 nm ($n = 43$)] with apparently three elongated spherical structures [length, 13.2 ± 1.3 nm ($n = 43$); width, 6.9 ± 0.7 nm ($n = 48$)] loosely associated with the central regions of the tail fibre. This unique base-plate structures resembles a rosette with three leaves, and can be observed in all five micrographs

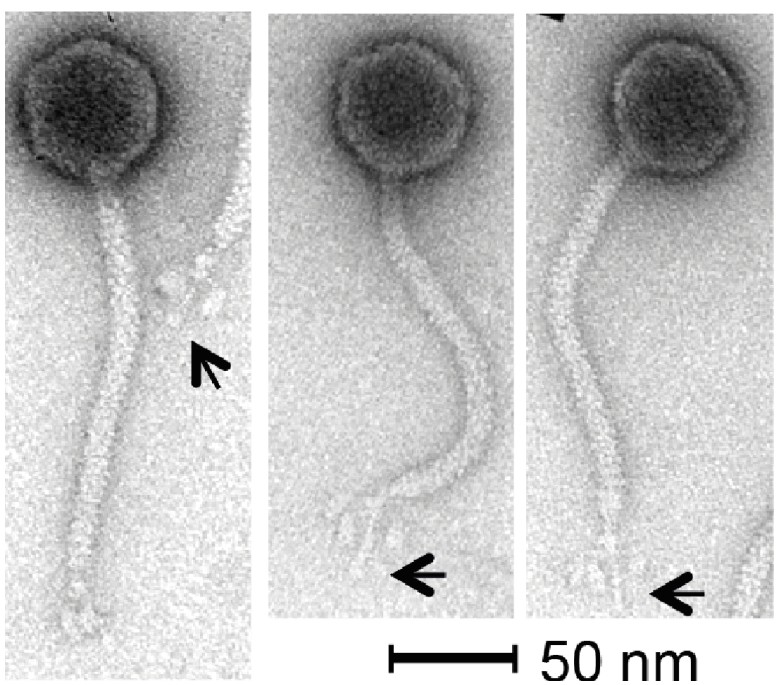

**Figure 2 Transmission electron micrographs of phage KLPN1.** The phage belongs to the family *Siphoviridae*, and has an unusual base-plate structure, resembling a rosette. The arrows indicate the central tail fibre surrounded by three flexible spherical base-plate structures.

shown in Fig. 2. A similar structure has been described for Rtp, a lytic phage of *Escherichia coli* (*Wietzorrek et al., 2006*), with which KLPN1 shares some sequence identity (discussed below). It is also visible in electron micrographs of phage φ28, which infects *K. pneumoniae* capsular type K28 (*Geyer et al., 1983*).

It was straightforward to isolate high-quality DNA from phage KLPN1. Heating an aliquot of CsCl-purified, DNAse-/RNAse-treated sample at 80 °C for 10 min followed by phenol/chloroform/isoamyl alcohol extractions and ethanol precipitation gave large quantities of high-quality DNA that was used for restriction digests (not shown) and genome sequencing. The estimated genome size of KLPN1 based on restriction digests with EcoRV or HaeIII was ∼45 kbp, which was an underestimation of the genome size of 49,037 bp as determined by sequencing (see below).

## Sequence analysis of KLPN1

The genome of KLPN1 was 49,037 bp with a G + C content of 50.53%, similar to previously sequenced *Klebsiella Siphoviridae* phages such as KP36 (Table 2). Initial genome annotation was performed using Genemark (*Besemer & Borodovsky, 1999*), from which a gff file was generated to allow visualisation of the predicted ORFs in Artemis v10.0 (*Rutherford et al., 2000*). Each ORF with a minimum amino acid content of 30, a start and stop codon as well as a ribosomal-binding site was retained (Table 3). The genome of KLPN1 was predicted to encompass 73 ORFs divided into four clusters, two rightward and two leftward (Fig. 3, Table 3). Of the 73 predicted ORFs, 23 were assigned a function with

**Table 2** Characteristics of previously sequenced *Klebsiella* phages.

| Phage | Family | Size (bp) | G+C(%) | No. of predicted ORFs | GenBank accession no. | Reference |
|---|---|---|---|---|---|---|
| NTUH-K2044-K1-1 | *Podoviridae* | 43,871 | 54.2 | 35 | NC_025418 | *Lin et al. (2014a)* |
| F19 | *Podoviridae* | 43,766 | 53.8 | 51 | NC_023567 | – |
| K11 | *Podoviridae* | 41,181 | 53.2 | 51 | NC_011043 | – |
| KP34 | *Podoviridae* | 43,809 | 54.1 | 57 | NC_013649 | *Drulis-Kawa et al. (2011)* |
| KP32 | *Podoviridae* | 41,119 | 52.4 | 44 | NC_013647 | *Kęsik-Szeloch et al. (2013)* |
| P13 | *Podoviridae* | 45,976 | 51.7 | 50 | – | *Shang et al. (2015)* |
| 0507-KN2-1 | *Myoviridae* | 159,991 | 46.7 | 154 | NC_022343 | *Hsu et al. (2013)* |
| JD001 | *Myoviridae* | 48,814 | 48.5 | 68 | NC_020204 | *Cui et al. (2012)* |
| KP27 | *Myoviridae* | 174,413 | 41.8 | 276 | NC_020080 | *Kęsik-Szeloch et al. (2013)* |
| KP15 | *Myoviridae* | 174,436 | 41.8 | 258 | NC_014036 | *Kęsik-Szeloch et al. (2013)* |
| vB_KleM-RaK2 | *Myoviridae* | 345,809 | 32 | 534 | NC_019526 | *Šimoliūnas et al. (2012)* |
| K64-1 | *Myoviridae* | 346,602 | 31.72 | 64 | AB897757 | *Pan et al. (2015)* |
| KP36 | *Siphoviridae* | 49,820 | 50.7 | 80 | NC_019781 | *Kęsik-Szeloch et al. (2013)* |
| phiKO2 | *Siphoviridae* | 51,601 | 51.5 | 64 | NC_005857 | *Casjens et al. (2004)* |
| 1513 | *Siphoviridae* | 49,462 | 50.61 | 72 | KP658157 | *Cao et al. (2015)* |
| KLPN1 | *Siphoviridae* | 49,037 | 50.5 | 73 | KR262148 | This study |

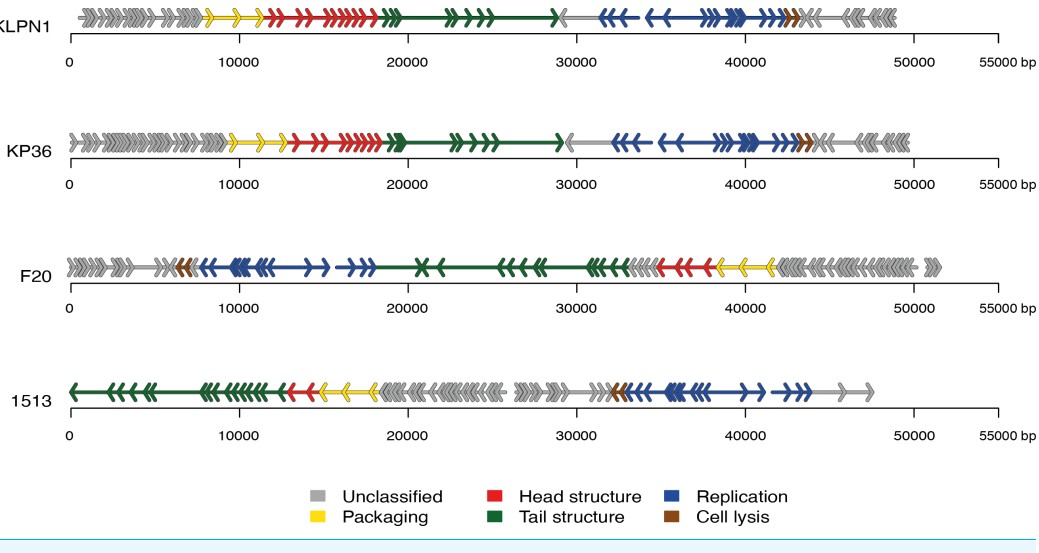

**Figure 3** Genome structures of the four members of the genus "Kp36likevirus". Phages KLPN1, KP36 and 1513 are virulent to *K. pneumoniae*, while F20 is virulent to *Enterobacter aerogenes* (*Mishra, Choi & Kang, 2012*).

the remainder representing hypothetical proteins with no assignable function (Table 3). BLAST analysis of the complete nucleotide sequence indicated that KLPN1 is closely related to the *Klebsiella* phages KP36 (95% identity across 86% of the genome; *Kęsik-Szeloch et al., 2013*), F20 (84% identity across 82% of the genome, GenBank accession no. JN672684; *Mishra, Choi & Kang, 2012*) and phage 1513 (95% identity across 85% of the genome, GenBank accession no. KP658157; *Cao et al., 2015*). Furthermore, partial identity was observed against *Shigella* phage Shfl1 as well as enterobacterial phages T1 (*Roberts,*

Hoyles et al. (2015), *PeerJ*, DOI 10.7717/peerj.1061

**Table 3** Genomic structure of phage KLPN1.

| KLPN1 genome | Start | Stop | Ribosome-binding site | Start | Strand | Predicted product[*] | Representative ORF in phage (% amino acid identity) | | | |
|---|---|---|---|---|---|---|---|---|---|---|
| | | | | | | | KP36 | F20 | 1513 | phiKO2 |
| KLPN1_01 | 516 | 1079 | AAGGAG | ATG | + | Unknown | – | – | – | – |
| KLPN1_02 | 1069 | 1290 | AGGAGG | ATG | + | HP | – | – | – | – |
| KLPN1_03 | 1395 | 1547 | GAGGAA | ATG | + | HP pSf1_0028 (*Shigella* phage pSf-1) | – | – | 39 (94) | – |
| KLPN1_04 | 1752 | 1997 | GAAGAtAG | TTG | + | HP (*Escherichia* phage e4/1c) | – | – | 38 (98.36) | – |
| KLPN1_05 | 2186 | 2416 | AGGAGGA | ATG | + | HP KP36_04 KP36 | 04 (74.65) | 76 (63.38) | 37 (92.11) | – |
| KLPN1_06 | 2487 | 2699 | AGAGGA | ATG | + | HP KP36_06 KP36 | 06 (82.86) | – | 36 (90) | – |
| KLPN1_07 | 2710 | 3030 | AAGGAGAA | ATG | + | HP KP36_11 KP36 | 11 (78.3) | 70 (65.09) | 35 (82.08) | – |
| KLPN1_08 | 3100 | 3471 | GGAGA | ATG | + | HP (Enterobacteria phage F20) | – | 69 (83.74) | – | – |
| KLPN1_09 | 3481 | 3783 | GAGGtGGccG | ATG | + | HP KP36_15 KP36 | 15 (91) | 68 (56) | 33 (85) | – |
| KLPN1_10 | 3783 | 4028 | AAcGGGA | ATG | + | HP KP36_16 KP36 | 16 (97.53) | 67 (83.95) | 32 (92.59) | – |
| KLPN1_11 | 4025 | 4210 | GAGGtAAAAG | ATG | + | HP KP36_17 KP36 (signal peptide) | 17 (85.25) | 66 (78.69) | 31 (58.18) | – |
| KLPN1_12 | 4291 | 4500 | AAtcAGGAG | ATG | + | HP (Enterobacteria phage F20) | – | 65 (73.91) | – | – |
| KLPN1_13 | 4500 | 4934 | GAGGAGA | ATG | + | HP (Enterobacteria phage F20) | 19 (33.93) | 64 (32.54) | 29 (35.58) | – |
| KLPN1_14 | 5004 | 5750 | AAGAGGAA | ATG | + | HP (Enterobacteria phage F20) | 20 (50.94) | 63 (54.55) | 28 (34.84) | – |
| KLPN1_15 | 5827 | 6099 | AGGAG | ATG | + | HP KP36_22 KP36 | 22 (65.56) | 61 (66.67) | – | – |
| KLPN1_16 | 6096 | 6668 | AAGGtGAtcGAA | ATG | + | HP KP36_23 KP36 | 23 (96.32) | 60 (78.53) | 25 (76.44) | – |
| KLPN1_17 | 6738 | 6974 | AAGGAG | ATG | + | HP KP36_24 KP36 | 24 (92.31) | 59a (86.3) | 24 (96.15) | – |
| KLPN1_18 | 6962 | 7195 | GGAGAAGG | ATG | + | HP KP36_25 KP36 | 25 (100) | 59 (87.01) | 23 (95.45) | – |
| KLPN1_19 | 7324 | 7518 | AAAGGG | ATG | + | HP KP36_26 KP36 | 26 (96.67) | 57 (93.65) | 22 (96.88) | – |
| KLPN1_20 | 7512 | 7718 | AAGGGAA | ATG | + | HP KP36_27 KP36 (signal peptide) | 27 (96.2) | 56 (64.81) | 21 (100) | – |
| KLPN1_21 | 7773 | 8378 | GAAGcGGcGtAA | ATG | + | terminase small subunit KP36 | 28(92.53) | 55 (91.95) | 20 (90.8) | – |
| KLPN1_22 | 8388 | 9992 | AAGGGccGA | ATG | + | terminase large subunit KP36 (terminase-like family: PF03237) | 29 (99.25) | 54 (98.13) | 19 (99.44) | – |
| KLPN1_23 | 10038 | 11348 | AAAcAAGGttGA | ATG | + | portal protein KP36 (phage portal protein, SPP1 Gp6-like: PF05133) | 30 (98.62) | 53 (91.78) | 18 (99.31) | – |
| KLPN1_24 | 11335 | 12102 | GGAGGcGGA | ATG | + | capsid morphogenesis protein KP36 (phage head morphogenesis domain: IPR006528) | 31 (96.44) | 52 (90.12) | 17 (95.22) | – |
| KLPN1_25 | 12104 | 12589 | GAGGttAG | ATG | + | HP gp40 (*Escherichia* phage phiEB49) HNH endonuclease (HNH endonuclease: PF13392) | – | – | – | – |
| KLPN1_26 | 12586 | 13716 | GGAGA | ATG | + | capsid protein KP36 | 32 (99.47) | 51 (93.09) | 16 (99.47) | – |
| KLPN1_27 | 13768 | 14286 | AGGAGA | ATG | + | HP KP36_33 KP36 | 33 (96.2) | 50 (50.87) | 15 (96.84) | – |
| KLPN1_28 | 14400 | 15359 | AGGAG | ATG | + | major capsid protein KP36_34 KP36 | 34 (98.75) | 49 (94.36) | 14 (98.43) | – |
| KLPN1_29 | 15451 | 15756 | GGAGcA | ATG | + | HP KP36_35 KP36 | 35 (100) | 48 (86.15) | 13 (98.77) | – |
| KLPN1_30 | 15753 | 16226 | AAAAGcGcGGA | ATG | + | HP KP36_36 KP36 | 36 (100) | 47 (92.75) | 12 (97.83) | – |
| KLPN1_31 | 16232 | 16609 | GGtAGGtGA | ATG | + | HP KP36_37 KP36 | 37 (99.2) | 46 (94.4) | 11 (98.4) | – |
| KLPN1_32 | 16602 | 17039 | AGGGtGGcGA | ATG | + | HP KP36_38 KP36 | 38 (100) | 45 (92.41) | 10 (99.05) | – |
Table 3 (*continued*)

| KLPN1 genome | Start | Stop | Ribosome-binding site | Start | Strand | Predicted product[*] | Representative ORF in phage (% amino acid identity) | | | |
|---|---|---|---|---|---|---|---|---|---|---|
| | | | | | | | KP36 | F20 | 1513 | phiKO2 |
| KLPN1_33 | 17029 | 17463 | GAGcGAGG | ATG | + | HP (Enterobacteria phage F20) Tail protein (phage tail protein: PF13554) | 39 (76.39) | 44 (88.19) | 09 (74.19) | – |
| KLPN1_34 | 17450 | 18136 | GGAGGcGA | ATG | + | HP (Enterobacteria phage F20) Tail protein (phage tail protein: PF08813) | – | 43/42 (68.26/42.04) | 08 (43.36) | – |
| KLPN1_35 | 18189 | 18845 | AGGAG | ATG | + | HP (Enterobacteria phage F20) Tail protein (phage tail protein: PF08813) | 40 (91.67) | 42 (82.87) | 08 (96.33) | – |
| KLPN1_36 | 18922 | 19248 | AGGAG | ATG | + | Tape measure chaperone protein (domain of unknown function DUF1789: PF08748) | 41 (98.15) | 41 (57.01) | 07 (98.15) | – |
| KLPN1_36.1 | 19239 | 19562 | – | – | + | Tape measure chaperone protein (phage-related hypothetical protein DUF1799: PF08809) | – | 40 (92.73) | 06 (96.36) | – |
| KLPN1_38 | 19620 | 22565 | GAtAAAGtAG | TTG | + | tail length tape-measure protein KP36 (λ phage tail tape-measure protein: PF09718) | 43 (91.63) | 39 (80.22/68.85) | 05 (91.04) | – |
| KLPN1_39 | 22568 | 22912 | AAcGAGGG | GTG | + | minor tail protein KP36 (minor tail protein: PF05939) | 44 (88.6) | 38 (87.72) | 04 (86.84) | 16 (28.32) |
| KLPN1_40 | 22949 | 23734 | – | | + | minor tail protein KP36 (minor tail protein L: TIGR01600) | 45 (93.17) | 37 (91.27) | 03 (92.77) | 17 (36.78) |
| KLPN1_41 | 23736 | 24473 | GAAAAGcGGAcGG | ATG | + | minor tail protein KP36 (endopeptidase, NlpC/P60 family: PF00877) | 46 (93.47) | 36 (96.33) | 02 (94.69) | 18 (42.62) |
| KLPN1_42 | 24448 | 25050 | AAGGA | ATG | + | tail assembly protein KP36 (λ tail assembly protein I: PF06805) | 47 (94.5) | 35 (98) | 01 (95.03) | – |
| KLPN1_43 | 25138 | 28839 | AGGAGG | ATG | + | tail fiber protein KP36 (putative phage tail protein: PF13550) | 48 (95.86) | 34 (88.6) | 01a (95.06) | 21 (39.75) |
| KLPN1_44 | 29027 | 31297 | AAGAGG | ATG | – | HP L418_01651 (*Klebsiellapneumoniae* UCICRE 7) (galactose-binding domain-like: IPR008979) | 49 (66.79) | 32 (83.49) | 71 (70.15) | – |
| KLPN1_45 | 31392 | 31853 | AGGAAcGA | ATG | – | single-stranded DNA binding protein KP36 (nucleic-acid binding proteins: SSF50249) | 50 (98.04) | 31 (82.35) | 70 (96.08) | – |
| KLPN1_46 | 31890 | 32546 | AAGGAAA | ATG | – | putative recombination protein KP36 (signal peptide) | 51 (99.54) | 30 (93.12) | 69 (98.62) | – |
| KLPN1_47 | 32606 | 33652 | GGAGcAA | ATG | – | exodeoxyribonuclease VIII KP36 (PD-(D/E)XK nuclease superfamily: PF12705) | 52 (97.7) | 29 (93.39) | 68 (97.99) | – |
| KLPN1_48 | 34148 | 35107 | GGAGGtAA | GTG | – | DNA primase KP36 (bacteriophage T7, Gp4, DNA primase/helicase, N-terminal: IPR013237) | 53 (98.7) | 28 (88.64) | 67 (99.35) | – |
| KLPN1_49 | 35183 | 35584 | GAGGGttAA | ATG | – | putative transcriptional regulator KP36 (λ repressor-like, DNA-binding domain: IPR010982) | 54 (98.5) | 27 (95.49) | 66 (99.25) | – |
| KLPN1_50 | 35676 | 37712 | AGGAttG | ATG | + | DNA helicase KP36 (helicase, C-terminal: IPR001650) | 55 (98.52) | 26 (94.25) | 65 (98.82) | – |
| KLPN1_51 | 37709 | 38116 | GGAGGcGAGG | GTG | + | HP KP36_56 KP36 (VRR-NUC domain: IPR014883) | 56 (99.08) | 25 (86.61) | 64 (100) | – |
| KLPN1_52 | 38181 | 38468 | GAAGAAcGGA | ATG | + | HP KP36_57 KP36 (signal peptide) | 57 (95.65) | – | 63 (94.87) | – |

Hoyles et al. (2015), *PeerJ*, DOI 10.7717/peerj.1061

Table 3 (*continued*)

| KLPN1 genome | Start | Stop | Ribosome-binding site | Start | Strand | Predicted product[*] | Representative ORF in phage (% amino acid identity) | | | |
|---|---|---|---|---|---|---|---|---|---|---|
| | | | | | | | KP36 | F20 | 1513 | phiKO2 |
| KLPN1_53 | 38471 | 39202 | GcGAGGttAA | ATG | + | DNA adenine methyltransferase KP36 (phage *N*-6-adenine-methyltransferase: TIGR01712) | 58 (97.53) | 23 (92.59) | 62 (97.94) | – |
| KLPN1_54 | 39204 | 39440 | GcGtAtGcGAA | ATG | + | HP KP36_59 KP36 | 59 (88.46) | 22 (79.49) | 61 (89.74) | – |
| KLPN1_55 | 39451 | 39741 | – | ATG | + | HP (Enterobacteria phage F20) | 60 (98.72) | 21 (83.33) | 60 (98.72) | – |
| KLPN1_56 | 39741 | 39989 | GGtGAcGA | ATG | + | HP KP36_61 KP36 | 61 (95.18) | 20 (89.02) | 59 (96.39) | – |
| KLPN1_57 | 40085 | 41215 | AAttGGGAtAA | ATG | + | HP KP36_62 KP36 (metallo-dependent phosphatase-like: IPR029052) | 62 (99.2) | 19 (94.15) | 58 (99.2) | – |
| KLPN1_58 | 41254 | 41745 | AAGGAAA | ATG | + | 3'-phosphatase, 5'-polynucleotide kinase KP36 (HAD superfamily, subfamily IIIB (acid phosphatase): PF03767) | 63 (93.87) | 18 (75.46) | 57 (95.09) | – |
| KLPN1_59 | 41742 | 42323 | GGAGtAGA | ATG | + | HP KP36_64 KP36 (P-loop containing nucleoside triphosphate hydrolase: IPR027417) | 64 (99.48) | 17 (88.17) | 56 (99.48) | – |
| KLPN1_60 | 42450 | 42665 | AGAGG | ATG | + | holin KP36 | 65 (83.1) | 16 (80.28) | 55 (83.1) | – |
| KLPN1_61 | 42667 | 43149 | AGGAGcAAG | ATG | + | endolysin KP36 (lysozyme-like domain: IPR023346) | 66 (86.23) | 15 (90.57) | 54 (98.12) | – |
| KLPN1_62 | 43146 | 43571 | AAGGA | ATG | + | Rz1A protein KP36 | 67 (96.45) | 14 (89.36) | 53 (99.29) | – |
| KLPN1_63 | 43641 | 44102 | GAGGtAA | ATG | – | HP KP36_68 KP36 | 68 (98.69) | 13 (88.89) | 52 (98.69) | – |
| KLPN1_64 | 44106 | 45674 | AGGAGcAAGG | ATG | – | HP KP36_69 KP36 (protein of unknown function DUF3987: PF13148) | 69 (99.81) | 12 (95.79) | 51 (99.81) | – |
| KLPN1_65 | 45800 | 46237 | GGAGAAAG | ATG | – | HP KP36_70 KP36 | 70 (97.93) | 11 (91.03) | 50 (98.62) | – |
| KLPN1_66 | 46238 | 46420 | GAAGAAA | ATG | – | HP KP36_71 KP36 | 71 (93.33) | 10 (86.44) | 49 (93.33) | – |
| KLPN1_67 | 46417 | 46620 | GGAGtAAAcGGA | ATG | – | HP KP36_72 KP36 | 72 (97.01) | 09 (65.67) | 48 (92.54) | – |
| KLPN1_68 | 46693 | 47385 | AAAtGGtGGA | ATG | – | HP KP36_73 KP36 | 73 (97.83) | 08 (90) | 47 (96.96) | 42 (56.83) |
| KLPN1_69 | 47382 | 47681 | GGcAtAG | TTG | – | HP KP36_74 KP36 | 74 (100) | 06 (55.41) | 46 (52.7) | – |
| KLPN1_70 | 47688 | 48065 | GAAAcGAGG | ATG | – | HP KP36_75 KP36 | 75 (96) | 05 (66.13) | 45 (70) | – |
| KLPN1_71 | 48065 | 48304 | GAGAAGGG | ATG | – | HP KP36_76 KP36 | 76 (96.2) | 04 (93.1) | 44 (97.47) | – |
| KLPN1_72 | 48301 | 48495 | AGGAGAA | ATG | – | HP KP36_77 KP36 | 77 (93.75) | 03 (84.81) | 43 (95.31) | – |
| KLPN1_73 | 48568 | 48849 | AGAGGG | ATG | – | HP KP36_78 KP36 | 78 (97.47) | 02 (84.81) | 42 (96.2) | – |

**Notes.**

[*] HP, hypothetical protein. Underlined predicted products are from InterProScan (http://www.ebi.ac.uk/interpro/) searches that returned results with the amino acid sequence encoded by each ORF (Table S2).

*Martin & Kropinski, 2004*) and RTP (*Wietzorrek et al., 2006*) (GenBank accession numbers HM035024, AY216660 and AM156909, respectively). A comparative analysis of KLPN1 was performed against both known members (KP36, F20) of the genus "Kp36likevirus" (*Niu et al., 2014*; Table 3, Fig. 3). For completion, KLPN1 was also compared with previously sequenced *Klebsiella* phages (Table 2). KLPN1 was shown not to share detectable homology with *Klebsiella* phages belonging to the family *Podoviridae* or *Myoviridae*. Based on the amino acid percentage identities presented in Table 3, the genomes of KLPN1 and related phages were divided into functional conserved modules: packaging, phage particle morphogenesis and DNA replication (Fig. 3, Table 3). Sections of the KLPN1 genome were shown to exhibit homology to the *Siphoviridae* phage phiKO2 genome, although such homology was confined to the tail morphogenesis region and one hypothetical protein (Table 3). Of the 73 predicted ORFs, only three appeared unique to KLPN1: two genes encoding hypothetical proteins located at the 5' end of the genome and a putative homing endonuclease-gene positioned in the capsid morphogenesis module (Table 3).

Based on sequence homology to ORF33 (encoding the host-specificity J protein; InterPro IPR021034) of phage T1, the phage receptor-binding protein of KLPN1 was predicted to be encoded by ORF43. The deduced ORF43 protein was found to exhibit similarity to fibronectin type III domain-containing protein, and potentially encodes the depolymerase whose effect was observed on the phage assay plates (Fig. 1). However, a HHpred search with the protein sequence encoded by this ORF failed to detect any domains associated with enzymes (such as glycanases, deacetylases and lyases) associated with depolymerase activity of *Klebsiella* lytic phages (*Geyer et al., 1983*). The protein encoded by ORF34 of phage NTUH-K2044-K1-1 has proven depolymerase activity associated with pectate lyase (*Lin et al., 2014a*); a HHpred search confirmed the protein encoded by this ORF has a lyase/polygalacturonase domain (1ru4_A pectate lyase, 94.3% probability of true positive, E-value 0.055, P-value $1.6 \times 10^{-6}$; 1bhe_A PEHA, polygalacturonase, 90.1% probability of true positive, E-value 13, P-value 0.00038). ORF96 of phage 0507-KN2-1 also encodes proven depolymerase activity (*Hsu et al., 2013*). A HHpred search demonstrated the protein encoded by this ORF has an acetylneuraminidase domain (3gw6_A endo-*N*-acetylneuraminidase, 99.8% probability of true positive, E-value $1 \times 10^{-19}$, P-value $2.9 \times 10^{-24}$). The whole-genome sequence of phage P13 cannot be retrieved from GenBank; however, on the basis of a PSI-BLAST search conducted by *Shang et al. (2015)*, genes 49 and 50 of phage P13 are predicted to encode the exopolysaccharide depolymerase, but no further substrate/functional information is available. HHpred searches of all proteins encoded by KLPN1 showed that ORF34 and ORF35 encode an endo-*N*-acetylneuraminidase/endosialidase domain (Table 4). Consequently, we predict ORF34 and/or ORF35 encode the depolymerase activity of phage KLPN1. Further work will be required to confirm our hypothesis.

The proteins encoded by ORF36 and ORF36.1 are predicted to be two chaperones, analogous to gpG and gpGT required for phage λ tail assembly (*Xu, Hendrix & Duda, 2013*). These ORFs are believed to keep the tail tape measure protein (TMP) soluble and to recruit the major tail protein subunits, both essential processes for tail assembly. ORF36

**Table 4 Selected HHpred results for ORF34 and ORF35 of phage KLPN1.**

| Hit | Probability of true positive | E-value | P-value | Score | SS | Cols | ORF amino acids | Reference |
|---|---|---|---|---|---|---|---|---|
| **ORF34** | | | | | | | | |
| 2k4q_A Major tail protein V; GPV, bacteriophage lambda, viral protein; NMR {Enterobacteria phage lambda} | 99.9 | $1.3 \times 10^{-27}$ | $3.6 \times 10^{-32}$ | 195.4 | 7.6 | 124 | 99-224 | *Pell et al. (2009)* |
| 3ju4_A Endo-*N*-acetylneuraminidase; endonf, polysia, high-resolution, glycosidase, hydrolase; HET: SLB; 0.98A {Enterobacteria phage K1F} | 81.6 | 1.4 | $4.1 \times 10^{-5}$ | 42.0 | 4.1 | 93 | 23-123 | *Schulz et al. (2010)* |
| 4hiz_A Endosialidase, PHI92_GP143; sialidase fold, beta-helix, endo-alpha2,8-sialidase, endo-Al sialidase sialic acid polymer; HET: SLB SIA SUC; 1.60A {Enterobacteria phage PHI92} | 80.9 | 4 | 0.00012 | 39.1 | 6.8 | 65 | 23-95 | Unpublished |
| **ORF35** | | | | | | | | |
| 2k4q_A Major tail protein V; GPV, bacteriophage lambda, viral protein; NMR {Enterobacteria phage lambda} | 99.9 | $5.3 \times 10^{-28}$ | $1.5 \times 10^{-32}$ | 195.1 | 8.6 | 119 | 94-214 | *Pell et al. (2009)* |
| 4hiz_A Endosialidase, PHI92_GP143; sialidase fold, beta-helix, endo-alpha2,8-sialidase, endo-Al sialidase sialic acid polymer; HET: SLB SIA SUC; 1.60A {Enterobacteria phage PHI92} | 91.4 | 0.55 | $1.6 \times 10^{-5}$ | 44.1 | 7.3 | 65 | 23-87 | Unpublished |
| 3ju4_A Endo-*N*-acetylneuraminidase; endonf, polysia, high-resolution, glycosidase, hydrolase; HET: SLB; 0.98A {Enterobacteria phage K1F} | 90.0 | 0.59 | $1.7 \times 10^{-5}$ | 43.8 | 6.2 | 65 | 23-87 | *Schulz et al. (2010)* |

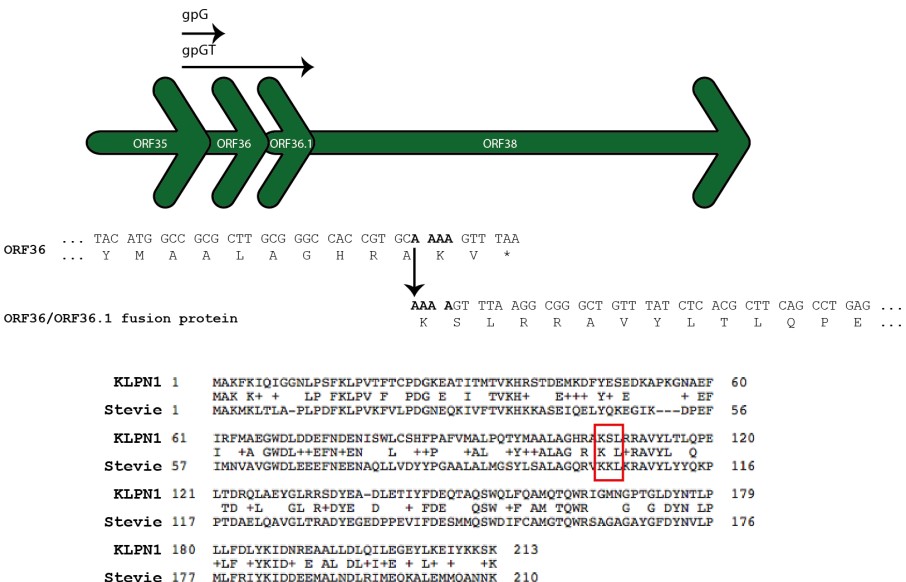

**Figure 4  ORF36 and ORF36.1 are related by a programmed-1 translational frameshift.** The frameshift allows translation of a gpGT-like protein (*Xu, Hendrix & Duda, 2013*). The ORF36/ORF36.1 fusion protein is predicted to contain almost the entire sequence of ORF36. *Citrobacter Siphoviridae* phage Stevie (GenBank accession number KM236241) has a conserved translational frameshift in its tape measure chaperone (GenBank accession number AIX12284) (*Shaw et al., 2015*).

encodes the short chaperone (gpG), while a programmed -1 translational frameshift facilitates a translational fusion between the products of ORF36 and ORF36.1 so as to form the long chaperone (gpGT) (hence the lack of a ribosome-binding site for ORF36.1) (Fig. 4). A BLASTP search of the predicted 213 aa fusion protein showed it shares 47% identity (E-value $3 \times 10^{-56}$) with the tape measure chaperone of the *Citrobacter Siphoviridae* phage Stevie, reported to have a conserved translational frameshift (*Shaw et al., 2015*; Fig. 4).

The genome sequence of KLPN1 was subjected to BLASTN searches against all publicly available virome sequence data held by METAVIR (51,992,208 sequence reads associated with 70 projects from a range of habitats, including human faeces (*Reyes et al., 2010*; *Kim et al., 2011*; *Minot et al., 2011*; *Minot et al., 2013*); Table S1). No hits corresponding to KLPN1 were found among these publicly available datasets. However, using InterProScan, the protein sequence encoded by ORF64 was found to belong to the family 'Protein of unknown function DUF3987', representing uncharacterized human-gut-microbiome-specific proteins (*Ellrott et al., 2010*; Table 3).

## DISCUSSION

*K. pneumoniae* subsp. *pneumoniae* is an enteric bacterium and important nosocomial and community-acquired opportunistic pathogen, causing pneumonia, and wound, burn, urinary tract and blood infections. There are 79 recognized capsular types of *K. pneumoniae* subsp. *pneumoniae*, with capsular types K1, K2, K5, K54 and K57 most frequently associated with invasive disease or pathogenicity; capsular type K20 is much rarer, though there is regional variability in the predominant clinically relevant capsular types (*Turton*

*et al., 2010*; *Pan et al., 2013*; *Hsu et al., 2013*). It has been suggested that the majority of *K. pneumoniae*-associated liver infections are preceded by colonization of the gastrointestinal tract, and one study has demonstrated familial spread of a virulent clone of *K. pneumoniae* causing liver disease (*Harada et al., 2011*; *Lin et al., 2012*). Using caecal effluent recovered from a healthy woman, we have isolated a capsular type K2 *rmpA*$^+$ strain (L4-FAA5) of *K. pneumoniae* subsp. *pneumoniae*. After K1, K2 strains are most frequently associated with pyogenic liver abscesses and frequently associated with community-acquired pneumonia (*Lin et al., 2014b*). K1 and K2 strains, along with the *rmpA* (regulator of mucoid phenotype A) gene, are associated with hypermucoviscosity and virulence. A study examining K1 and K2 strains of *K. pneumoniae* in 43 Taiwanese patients with liver abscesses showed, using pulsed-field gel electrophoresis (PFGE), that 17 randomly selected pairs of patient faecal or saliva and abscess isolates allowed differentiation of patients based on the PFGE profiles of their isolates. The abscess isolates had PFGE profiles identical, or closely related to, those of faecal or saliva isolates from the same patient (*Fung et al., 2012*). The similarity between each patient's faecal and abscess PFGE profiles led *Fung et al. (2012)* to suggest that the patients' infections arose from bacteria originating in the gut microbiota. Our isolation of a *K. pneumoniae* strain with virulence traits from the human caecum supports the assertion that the human gut microbiota is a source of potentially infectious *K. pneumoniae*. Limited data are available on the carriage and diversity of gut/faecal *K. pneumoniae*. A study on healthy Chinese adults in Chinese territories and overseas showed between 18.8 and 87.7% faecal carriage of *K. pneumoniae*, with individuals in Malaysia (64/73) and Taiwan (150/200) showing the highest carriage rates and those in Japan (6/32) showing the lowest (*Lin et al., 2012*). Isolates were tested with antisera for capsular types K1–K74 and K80–82: K1 and K2 isolates accounted for ∼10% of isolates in all countries, at least one representative of each capsular type was detected in the study and non-typable (11–88%) isolates were reported for 7/8 countries (*Lin et al., 2012*). The high carriage of *K. pneumoniae* in Taiwan is thought to contribute to the high incidence of liver abscess disease seen in the country (*Fung et al., 2012*).

As with other nosocomial opportunistic pathogens, broad-spectrum antibiotic resistance is a feature of *K. pneumoniae* subsp. *pneumoniae* and limits treatment options (*Cantón et al., 2012*). Therefore, identification of alternative treatment therapies or adjuncts to existing therapies for infections associated with this organism is of utmost importance. Phages against *K. pneumoniae* subsp. *pneumoniae* have been used to successfully treat *K. pneumoniae* infections in animal models of sepsis, pneumonia, burn wounds and liver disease, without causing apparent harm to animals (*Bogovazova, Voroshilova & Bondarenko, 1991*; *Malik & Chhibber, 2009*; *Chhibber, Kaur & Kumari, 2008*; *Kumari, Harjai & Chhibber, 2009*; *Kumari, Harjai & Chhibber, 2010b*; *Hung et al., 2011*). The number of known lytic phages that infect *K. pneumoniae* remains small and their range is limited to a small number of capsular types (though phenotypic information on strains is absent from the majority of publications on *K. pneumoniae* phages). Given the large number of phages present in faeces and caecal effluent (*Hoyles et al., 2014*), and because *K. pneumoniae* subsp. *pneumoniae* is a member of the human gut microbiota, we were keen to exploit this envi-

ronment as a source of phages with potential therapeutic applications. We isolated from caecal effluent a phage (named KLPN1) that infects *K. pneumoniae* subsp. *pneumoniae* L4-FAA5 and K2 clinical isolates of *K. pneumoniae* subsp. *pneumoniae* (Table 1). Phage KLPN1 does not infect non-K2 clinical isolates (Table 1). To the best of our knowledge, this is the first report of the isolation of a bacterium–phage combination from the human caecum.

To date, only eight phages infecting the K2 capsular type of *K. pneumoniae* have been reported (*Bogovazova, Voroshilova & Bondarenko, 1991*; *Malik & Chhibber, 2009*; *Kumari, Harjai & Chhibber, 2010b*; *Hung et al., 2011*). Several *Podoviridae* were isolated on *K. pneumoniae* B5055 (assumed to be derived from NCTC 5055) and tested as therapeutic agents [Kpn5, Kpn12, Kpn13, Kpn17, Kpn22 (individually and in cocktail) and KØ1] in B5055-induced burn-wound infections in mice (*Kumari, Harjai & Chhibber, 2009*; *Kumari, Harjai & Chhibber, 2010a*; *Kumari, Harjai & Chhibber, 2010b*; *Malik & Chhibber, 2009*). Phage SS (*Podoviridae*) was isolated on B5055 and used to treat lobar pneumonia caused by the same strain in mice (*Chhibber, Kaur & Kumari, 2008*). Phage $\phi$NK5 (*Podoviridae*) was isolated from sewage on *K. pneumoniae* NK-5, isolated from a patient with a primary liver abscess and septicaemia (*Hung et al., 2011*). The same bacterial strain was used to induce liver abscesses and bacteraemia in mice, which were successfully treated with $\phi$NK5. None of the aforementioned phages has been tested against a range of clinical *K. pneumoniae* isolates. Therefore, it is difficult to assess how useful they would be in treatment of a wide range of clinical infections, especially when similar infections can be caused by different capsular types of *K. pneumoniae* (e.g., Table 1). We can state that KLPN1 has potential for treating K2-associated infections, but note that differences in lytic infection were observed between the caecal isolate and the clinical isolates: KLPN1 infected the caecal isolate and exhibited depolymerase activity that was absent with the clinical K2 isolates. Further investigations may reveal that its infection kinetics differ between K2 strains.

*Hung et al. (2011)* demonstrated the generation of phage-insensitive mutants after 6 and 12 h co-cultures of NK-5/$\phi$ NK5, with these mutants lacking the hypermucoviscosity phenotype of NK-5. On solid media, phage-insensitive mutants of *K. pneumoniae* L4-FAA5 were routinely observed after 48 h incubation with phage KLPN1. Whether mutants are generated more quickly in liquid culture remains to be determined, as does the nature of the mutations that allow them to escape lytic infection with KLPN1. Does solid media act as a spatial refuge (*Mills et al., 2013*) for sensitive bacteria, delaying the appearance of insensitive mutants?

It is predicted that many members of the human gut microbiota are embedded in biofilms, and phages may contribute to cell lysis in these ecological niches (*Mills et al., 2013*). The microbiota of the human caecum resides in a highly mucoid biofilm (*Randal Bollinger et al., 2007*). Phage KLPN1 exhibits depolymerase activity on *K. pneumoniae* L4-FAA5, which may facilitate the movement of the phage within the caecal biofilm. It has been demonstrated that phages can diffuse within biofilms, be immobilized, amplified and released after a lytic cycle in these environments. They may also potentially interact with their specific binding sites on bacteria, even in the absence of lytic activity (*Mills et al., 2013*). Further work is needed to understand interactions between *K. pneumoniae*

L4-FAA5 and phage KLPN1 in single- and multi-strain biofilm systems, and to determine whether the "spatial refuge hypothesis" (*Mills et al., 2013*) holds for the caecal microbiota, preventing the extinction of all sensitive bacteria within the biofilm.

Following whole-genome sequencing and annotation, KLPN1 was found to encompass 73 ORFs and, based on its sequence homology to *Klebsiella* phages KP36 and F20, it is evident that KLPN1 is a member of the family *Siphoviridae*, subfamily *Tunavirinae*, and would become the third member of the genus "Kp36likevirus" (*Niu et al., 2014*). In addition, based on our sequence analyses phage 1513 (a *Siphoviridae* that infects a multidrug-resistant *K. pneumoniae* strain isolated from a patient with pneumonia; *Cao et al., 2015*) can also be added to this genus, bringing the total number of members to four. This classification would be consistent with the morphological appearance of KLPN1, which has a capsid of 64 nm as well as rosette-like or propeller tail tip, a finding also observed for phages $\phi$28, T1 and RTP (*Geyer et al., 1983*; *Wietzorrek et al., 2006*). Unsurprisingly, KLPN1 shares no sequence homology with *Klebsiella* phages belonging to the families *Myoviridae* or *Podoviridae*; however, some similarities to *Siphoviridae* phage phiKO2 were observed, but these were confined to the tail morphogenesis region and a hypothetical protein. ORF60 and ORF61 are predicted to encode holin (which destroys the cytoplasmic membrane) and endolysin (which degrades peptidoglycan), respectively. These gene products have antibacterial properties that can be used in phage-associated therapies (*Viertel, Ritter & Horz, 2014*); therefore, further characterization of the proteins encoded by these ORFs is required.

## CONCLUSIONS

We isolated a *K. pneumoniae* subsp. *pneumoniae*–phage combination from the human caecum, and have characterized its lytic properties against a panel of *K. pneumoniae* subsp. *pneumoniae* clinical isolates. Phage KLPN1 infects capsular type K2 isolates, and may have applications in treating a range of K2-associated infections. We have, therefore, demonstrated the gut microbiota as a source of clinically relevant phages. Whole-genome sequence analysis of KLPN1 revealed the phage to encode proteins that have potential applications in phage-associated therapies. Characterization of these gene products, or genetically modified variants, is required to determine their usefulness.

## ACKNOWLEDGEMENTS

HN and KJH acknowledge the technical assistance of Angela Back in sample preparation for electron microscopy.

### Funding

Lesley Hoyles held a Government of Ireland Postdoctoral Fellowship in Science, Engineering and Technology (IRCSET EMPOWER scheme). Douwe van Sinderen is a recipient of a Science Foundation Ireland (SFI) Principal Investigator award (Ref. No. 13/1A/1953). James Murphy is the recipient of an SFI-funded Technology Innovation Development

Award (TIDA) (Ref. No. 14/TIDA/2287). The funders had no role in study design, data collection and analysis, decision to publish, or preparation of the manuscript.

## Grant Disclosures

The following grant information was disclosed by the authors:
Government of Ireland Postdoctoral Fellowship in Science, Engineering and Technology.
Science Foundation Ireland (SFI) Principal Investigator award: 13/1A/1953.
SFI-funded Technology Innovation Development Award: 14/TIDA/2287.

## Competing Interests

The authors declare there are no competing interests.

## Author Contributions

- Lesley Hoyles and James Murphy conceived and designed the experiments, performed the experiments, analyzed the data, contributed reagents/materials/analysis tools, wrote the paper, prepared figures and/or tables, reviewed drafts of the paper.
- Horst Neve performed the experiments, analyzed the data, wrote the paper, prepared figures and/or tables, reviewed drafts of the paper.
- Knut J. Heller performed the experiments, wrote the paper, reviewed drafts of the paper.
- Jane F. Turton and Jeremy D. Sanderson performed the experiments, contributed reagents/materials/analysis tools, wrote the paper, reviewed drafts of the paper.
- Jennifer Mahony conceived and designed the experiments, performed the experiments, analyzed the data, wrote the paper, reviewed drafts of the paper.
- Barry Hudspith performed the experiments, wrote the paper, liaised with Dr Sanderson to arrange access to colonoscopy suite, delivered samples.
- Glenn R. Gibson contributed reagents/materials/analysis tools, wrote the paper, reviewed drafts of the paper.
- Anne L. McCartney performed the experiments, analyzed the data, contributed reagents/materials/analysis tools, wrote the paper, reviewed drafts of the paper.
- Douwe van Sinderen analyzed the data, contributed reagents/materials/analysis tools, wrote the paper, prepared figures and/or tables, reviewed drafts of the paper.

## Human Ethics

The following information was supplied relating to ethical approvals (i.e., approving body and any reference numbers):
Ethical approval to collect caecal effluent from patients was obtained from St Thomas' Hospital Research Ethics Committee (06/Q0702/74) covering Guy's and St Thomas' Hospitals, and transferred by agreement to London Bridge Hospital.

## DNA Deposition

The following information was supplied regarding the deposition of DNA sequences:
GenBank: accession number KR262148.

## Supplemental Information

Supplemental information for this article can be found online at http://dx.doi.org/10.7717/peerj.1061#supplemental-information.

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
