# Peer review of "Klebsiella pneumoniae subsp. pneumoniae–bacteriophage combination from the caecal effluent of a healthy woman"

_PeerJ, doi:10.7717/peerj.1061_

## Round 0.1 · original submission · Minor Revisions

This paper reports the characterization of a bacteriophage isolated from a Klebsiella pneumoniae subsp. pneumoniae obtained from a caecal effluent of a healthy woman, with the idea of better understand the gastrointestinal microbial ecology and the potential therapeutic use of phages to control important disease caused by bacterial pathogenic strains.

Both reviewers like the work and the writing. Authors should follow all the comments carefully.

Reviewer 1 ·

Basic reporting

This study reports the genome sequence and infection capacity of a new phage KLPN1 isolated from a colonoscopy of an healthy female. This phage was found to infect Klebsiella K2 strains, which have been associated with clinical infections.
Currently there is reduced information on phages inhabiting the human gut and these can be important not only for a better understanding of virus diversity within the gut, but also because they can be a source of future therapeutic treatments.
The study seems to me to be well conducted and the data solid.

Experimental design

I only have one minor comment regarding the sentence on Line409 - 410. Can the authors be a bit more precise here: what is the frequency at which insensitive mutants spontaneously arise?

Validity of the findings

Of course I think that further characterization of the phage isolated will be needed to evaluate its genomic content and therapeutic potential, but the authors acknowledge this and so I am happy with their discussion of the the results shown.

Additional comments

I found the paper to bring new and interesting results to the literature in this field. To the best of my knowledge the data presented is solid and the paper is clearly written. The introduction is put in a general context and the discussion points out important future work.

Reviewer 2 ·

Basic reporting

This paper reports the characterization of a bacteriophage isolated from a Klebsiella pneumoniae subsp. pneumoniae obtained from a caecal effluent of a healthy woman, with the idea of better understand the gastrointestinal microbial ecology and the potential therapeutic use of phages to control important disease caused by bacterial pathogenic strains.
This bacteriophage, named KLPN1, was morphologically characterized and classified as a member of the Kp36likevirus genus within the Siphoviridae family. It has the ability of specifically lysate Klebsiella capsular K2 strains, and contains a DNA genome of 40,037 bp in length, composed of 73 ORF encoding for proteins with known and unknown functions.
This is a very well written paper that contains important novel information about bacteriophages belonging to the Kp36likevirus genus, which can be used for the control of important infections caused by Klebsiela pneumoniae.

Experimental design

The methodology is described with sufficient information to be reproducible, and the experimental design used is adequate to support the results obtained.

Validity of the findings

The results obtained answer the research questions proposed.

Additional comments

Characterization of phage KLPN1 is conducted in a very nice way, and all the findings are very well supported. However, no background of phage 1513 was included, thus, information in this regard needs to be explained or supported with references.
It will be useful to include Information about the absence of KLPN1 depolymerase activity with any of the K2 isolates, stated in lines 250 and 251 in Table 2.

Figure 2 includes 5 nice images of KLPN1 that show the same characteristics. It could be consider showing only one of them.

---

## Round 0.2 · accepted · Accept

The paper is very nice and all the queries of reviewers were responded I see no impediment for publication